# p$K_a$ of the ligand water molecules in the oxygen-evolving Mn$_4$CaO$_5$ cluster in photosystem II

Keisuke Saito [1,2], Minesato Nakagawa[1] & Hiroshi Ishikita [1,2✉]

Release of the protons from the substrate water molecules is prerequisite for $O_2$ evolution in photosystem II (PSII). Proton-releasing water molecules with low p$K_a$ values at the catalytic moiety can be the substrate water molecules. In some studies, one of the ligand water molecules, W2, is regarded as $OH^-$. However, the PSII crystal structure shows neither proton acceptor nor proton-transfer pathway for W2, which is not consistent with the assumption of $W2 = OH^-$. Here we report the p$K_a$ values of the four ligand water molecules, W1 and W2 at Mn4 and W3 and W4 at $Ca^{2+}$, of the Mn$_4$CaO$_5$ cluster. p$K_a$(W1) ≈ p$K_a$(W2) << p$K_a$(W3) ≈ p$K_a$(W4) in the Mn$_4$CaO$_5$ cluster in water. However, p$K_a$(W1) ≈ p$K_a$(D1-Asp61) << p$K_a$(W2) in the PSII protein environment. These results suggest that in PSII, deprotonation of W2 is energetically disfavored as far as W1 exists.

[1] Department of Applied Chemistry, The University of Tokyo, 7-3-1 Hongo, Bunkyo-ku, Tokyo 113-8654, Japan. [2] Research Center for Advanced Science and Technology, The University of Tokyo, 4-6-1 Komaba, Meguro-ku, Tokyo 153-8904, Japan. ✉email: hiro@appchem.t.u-tokyo.ac.jp

In the water-splitting enzyme, photosystem II (PSII), oxygen evolution proceeds, removing four protons ($H^+$) and four electrons from two substrate water molecules at the oxygen-evolving complex, $Mn_4CaO_5$ (Fig. 1)[1,2]. The $Mn_4CaO_5$ cluster has two ligand water molecules, W1 and W2, at the dangling Mn4 site and another two ligand water molecules, W3 and W4, at the $Ca^{2+}$ site. These bound water molecules are candidates as potential substrates for water oxidation. As electron transfer occurs, the oxidation state of the oxygen-evolving complex, $S_n$, increases. Release of protons is observed with the typical stoichiometry of 1:0:1:2 for $S_0 \rightarrow S_1 \rightarrow S_2 \rightarrow S_3 \rightarrow S_0$, and $O_2$ is evolved in the $S_3$ to $S_0$ transition. After $O_2$ evolution, the first proton-releasing step is the $S_0$ to $S_1$ transition. The $Mn_4CaO_5$ cluster has a chain of strongly H-bonded 8 water molecules (O4-water chain) directly linked to O4 (linking Mn4 and Mn3 in the $Mn_3CaO_4$-cubane) and the release of the proton occurs along the O4-water chain in the $S_0$ to $S_1$ transition[3–5]. In $S_0$ and $S_1$[6], the ligand water molecules W1–W4 are $H_2O$ in quantum mechanical/molecular mechanical (QM/MM) models (i.e., in the presence of the PSII protein environment)[3,7,8], whereas W2 is assumed to be $OH^-$ in simplified QM models (i.e., in the absence of the PSII protein environment)[9–11]. The release of the proton is also observed in the $S_2$ to $S_3$ transition. Based on the observations of the recent radiation-damage-free structures obtained using the X-ray free electron laser (XFEL), the sixth O site, O6, may be incorporated into the Mn1 and O5 moieties of the $Mn_4CaO_5$ cluster in the $S_2$ to $S_3$ transition[12–14].

During the water incorporation process, deprotonation of the water molecule may occur as proposed (e.g., water incorporation into the Mn1 moiety[15,16], water incorporation from the $Ca^{2+}$ moiety[17–19]). In theoretical models by Shoji et al.[17] and Isobe et al.[20], it was assumed the presence of $OH^-$ at W2 in $S_2$ to facilitate the release of the proton from $H_2O$ at W3 to $OH^-$ at W2, and proposed that deprotonation of W3 at the $Ca^{2+}$ moiety occurs in the $S_2$ to $S_3$ transition. If $OH^-$ needed to be located at either W2 or W3, placing $OH^-$ at W2 and $H_2O$ at W3 would be consistent with the experimentally measured values, $pK_a(Ca^{2+})$ = 12.8 >> $pK_a(Mn^{3+})$ = 0.7[21] in water. In theoretical models by Ames et al.[22], Rapatskiy et al.[23], Pérez-Navarro et al.[24], and Capone et al.[25], W2 was also assumed to be $OH^-$.

If the $Mn_4CaO_5$ cluster were isolated from the protein environment, placing $OH^-$ at W2, not at W3, might be explained by $pK_a(Ca^{2+})$ >> $pK_a(Mn^{3+})$. However, placing $OH^-$ at W2 is not consistent with the PSII crystal structures. The PSII crystal structures show that W2 has no strong H-bond acceptor, whereas

W1 has a strong H-bond acceptor, D1-Asp61. Quantum mechanical/ molecular mechanical calculations show that $H_2O$ at W1 forms a low-barrier H-bond with D1-Asp61 and is ready for proton transfer in $S_2$[26]. This may correspond to the significant changes in the H-bond properties between D1-Asp61 and a water molecule in the $S_1$ to $S_2$ transition observed in Fourier transform infrared (FTIR) spectroscopy[27]. Consistently, FTIR spectra suggested that W2 is $H_2O$ in $S_1$ and $S_2$[8].

As far as we are aware, the $pK_a$ values of the four water molecules at the $Mn_4CaO_5$ moiety, even those for the ligand water molecules W1–W4 are not reported. Robertazzi et al. reported that in $S_1$, $pK_a(W2)$ = 6.1 for the isolated $Mn_4CaO_5$ cluster with deprotonated D1-His337 and 7.8 for the isolated $Mn_4CaO_5$ cluster with protonated D1-His337 in water, based on quantum chemical calculations[28]. However, the $pK_a$ values for W1, W3, and W4 are not reported, which prevent from identifying the deprotonation sites even in the isolated $Mn_4CaO_5$ cluster in water. It should also be noted that the definition of the $Mn_4CaO_5$ cluster is vague. It can be comprised of $Mn_4CaO_5$, four ligand water molecules (W1–W4), and seven ligand residues (D1-Asp170, D1-Glu189, D1-His332, D1-Glu333, D1-Asp342, D1-Ala344, and CP43-Glu354, Fig. 1). It can also include the second sphere ligand residues, D1-Asp61, and CP43-Arg357, or the O4-water chain that forms an H-bond with O4. D1-Asp61 serves as an H-bond acceptor for W1 and is likely to facilitate proton transfer in the $S_2$ to $S_3$ transition[26]. The O4-water chain forms a significantly short H-bond with O4 (O...O < 2.5 Å) in the crystal structures in $S_1$ (or a slightly lower S-state)[29,30] and facilitates the release of the proton from O4 in the $S_0$ to $S_1$ transition[3–5]. The involvement of these proton acceptor groups (i.e., proton transfer pathways) facilitates deprotonation of the H-bond donor sites of the $Mn_4CaO_5$ cluster and decreases the $pK_a$ values. This fact already implies that the $pK_a$ values of the isolated $Mn_4CaO_5$ cluster in water are far from the relevant $pK_a$ values in the PSII protein environment.

Here we report the $pK_a$ values of the W1–W4 sites in the isolated $Mn_4CaO_5$ cluster in water, using quantum chemical approaches. To investigate the energetics of release of the proton towards the proton-transfer pathways in PSII, we analyze the potential-energy profiles of the H-bonds between the ligand water molecules and the H-bond acceptor (i.e., the proton acceptor) groups in the PSII protein environment. The results show that $pK_a(W1) \approx pK_a(W2) << pK_a(W3) \approx pK_a(W4)$ in the $Mn_4CaO_5$ cluster in water (i.e., in the absence of the PSII protein environment), whereas $pK_a(W1) \approx pK_a(D1\text{-}Asp61) << pK_a(W2)$ in PSII.

## Results and discussion

We calculate the energy difference ($\Delta E_{water}$) between the protonated and deprotonated states of hexa-aqua metal complexes in water (Fig. 2). The calculated $\Delta E_{water}$ values of hexa-aqua metal complexes with the valences of II, III, and IV show a correlation with the experimentally measured $pK_a$ values (Fig. 3) and are best fitted to the following equation:

$$pK_a = 0.220 \, \Delta E_{water}[\text{kcal/mol}] - 55.8 \qquad (1)$$

The calculated $pK_a$ values of hexa-aqua metal complexes obtained using Eq. 1 are listed in Table 1. Note that in vacuum, the experimentally measured $pK_a$ values cannot be reproduced using a single equation (Supplementary Fig. 1, Supplementary Table 1), because the electrostatic influence between the cationic metal and anionic $OH^-$ in the deprotonated state is overestimated in vacuum. Thus, $pK_a$ predominantly depends on the metal valence in vacuum.

The isolated $Mn_4CaO_5$ cluster is comprised of $Mn_4CaO_5$, four ligand water molecules (W1–W4), and seven ligand residues (D1-

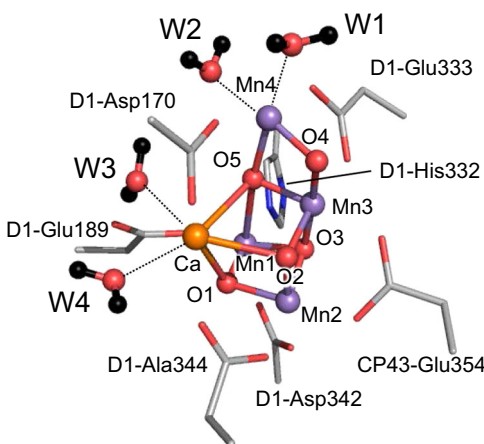

**Fig. 1 Structure of the $Mn_4CaO_5$ cluster.** The second sphere ligand residues (D1-Asp61 and CP43-Arg357) are not shown. Dotted lines indicate ligations of the ligand water molecules to the Mn4 and $Ca^{2+}$ sites.

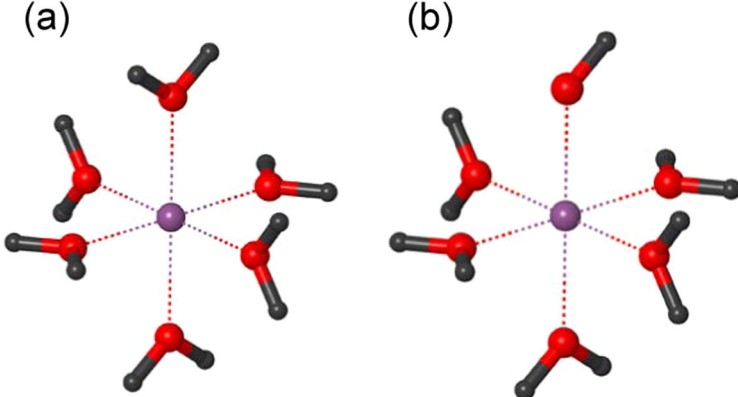

**Fig. 2 Structures of hexa-aqua metal complexes. a** Protonated state. **b** Deprotonated state. Magenta balls indicate metal ions. Dotted lines indicate ligations of the ligand water molecules to metal ions.

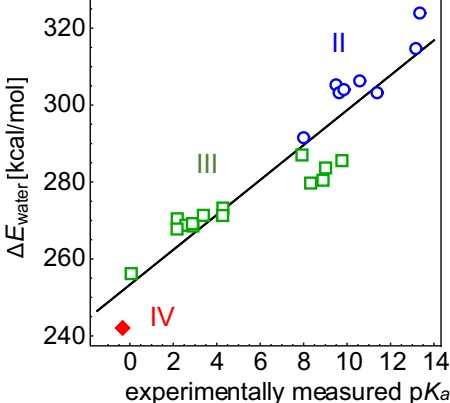

**Fig. 3 Experimentally measured p$K_a$ values and calculated H$_2$O/OH$^-$ energy differences of hexa-aqua metal complexes.** Blue open circles for divalent (II) metals, green open squares for trivalent (III) metals, and red closed diamond for the tetravalent (IV) metal.

**Table 1 Calculated (Calc.) and experimentally measured (Expl.) p$K_a$ of hexa-aqua metal complexes.**

| Metal ion | Calc. p$K_a$ | Expl. p$K_a$ |
|---|---|---|
| Zr$^{4+}$ | −2.6 | −0.32[a] |
| Mn$^{3+}$ (high spin) | 0.5 | 0.08[b] |
| Fe$^{3+}$ (high spin) | 3.1 | 2.19[b] |
| Ti$^{3+}$ | 3.7 | 2.20[b] |
| V$^{3+}$ | 3.3 | 2.60[b] |
| Ru$^{3+}$ (low spin) | 3.4 | 2.90[b] |
| Co$^{3+}$ (low spin) | 3.3 | 2.92[b] |
| Rh$^{3+}$ (low spin) | 3.9 | 3.40[b] |
| Cr$^{3+}$ | 3.9 | 4.29[b] |
| Sc$^{3+}$ | 4.3 | 4.30[b] |
| Lu$^{3+}$ | 5.7 | 7.94[a] |
| Cu$^{2+}$ | 10.9 | 8.0[b] |
| Y$^{3+}$ | 7.0 | 8.34[a] |
| Pr$^{3+}$ | 5.9 | 8.91[a] |
| La$^{3+}$ | 7.3 | 9.03[a] |
| Fe$^{2+}$ (high spin) | 11.3 | 9.50[b] |
| Co$^{2+}$ (high spin) | 10.9 | 9.65[b] |
| Gd$^{3+}$ | 6.6 | 9.78[a] |
| Ni$^{2+}$ | 11.1 | 9.86[b] |
| Mn$^{2+}$ (high spin) | 11.6 | 10.59[b] |
| Mg$^{2+}$ | 8.3 | 11.41[a] |
| Sr$^{2+}$ | 13.4 | 13.18[a] |
| Ba$^{2+}$ | 15.5 | 13.36[a] |
| RMSD[c] | 1.5 | |

[a]Ref. 21.
[b]Ref. 34.
[c]Root mean square deviation (RMSD) of the calculated p$K_a$ values from the experimentally measured p$K_a$ values.
Calc. p$K_a$ is obtained using Eq. (1).

Asp170, D1-Glu189, D1-His332, D1-Glu333, D1-Asp342, D1-Ala344, and CP43-Glu354, Fig. 1). Using Eq. (1), the p$K_a$ values for W1–W4 are calculated at the isolated Mn$_4$CaO$_5$ cluster in water (in the absence of the PSII protein environment, Fig. 1). p$K_a$(W1) and p$K_a$(W2) on the dangling Mn4 site are 7–11, whereas p$K_a$(W3) and p$K_a$(W4) on Ca$^{2+}$ site are 14–18 (Table 2). p$K_a$(W1) and p$K_a$(W2) are higher than p$K_a$ of 0.5 for Mn$^{3+}$ (Table 1), because Mn4 has two ligand acidic residues, D1-Asp170 and D1-Glu333, and two μ-oxo O atoms, O4 and O5 (Fig. 1). The difference in the p$K_a$ of >5 between the Mn4 and Ca$^{2+}$ sites (Table 2) indicate that the Ca$^{2+}$ site is originally disadvantageous for H$_2$O deprotonation with respect to the Mn4 site. In particular in the PSII protein environment, W1 at Mn4 has D1-Asp61 as an H-bond acceptor, whereas W3 and W4 at Ca$^{2+}$ do not have the corresponding acidic residues (see below). Thus, deprotonation of H$_2$O and incorporation of the generated OH$^-$ into the Mn$_4$CaO$_5$ cluster occurring at the Ca$^{2+}$ moiety in the S$_2$ to S$_3$ transition (e.g., refs. [17–19]) needs to overcome the energetic disadvantage.

As far as the ligand coordination in the PSII protein structure is maintained (i.e., the torsion angles are fixed), p$K_a$(W2) is only marginally (~1 p$K_a$ unit) lower than p$K_a$(W1) in the absence of the PSII protein environment (Table 2). When the geometry is fully relaxed (i.e., the torsion angles are not fixed) and OH$^-$ is initially placed at W4 [to calculate p$K_a$(W4)], proton transfer occurs from W2 via W3 to W4 occurs and OH$^-$ is finally stabilized at W2, not at W1 (Supplementary Fig. 2). Deprotonation of W2 instead of W1 is just an artifact as W2, W3, and W4 form

the H-bond network, pushing W1 and W3 away from Mn4 and Ca$^{2+}$, respectively (W1...Mn4 = 3.8 Å and W3...Ca$^{2+}$ = 3.6 Å). These observations may be a basis of why electron paramagnetic resonance (EPR) signals were often interpreted based on theoretical models, in which W2 was assumed to be OH$^-$ in QM-based models (i.e., in the absence of the PSII protein environment, e.g., refs. [10,22,23]). Interestingly, W2 = OH$^-$ are also assumed in other QM-based models (e.g., by Siegbahn[9] and Retegan et al.[11]), without using QM/MM approaches. It should be noted that W2, W3, and W4 never form the H-bond network as far as the PSII protein environment exists.

The marginally low p$K_a$(W2) with respect to p$K_a$(W1) (Table 2) were the case only when the Mn$_4$CaO$_5$ cluster could be ideally isolated from the PSII protein environment. In such a model

system, $pK_a$(W1) and $pK_a$(W2) can change easily, depending on even the definition of the $Mn_4CaO_5$ cluster. When the second sphere ligand residues (D1-Asp61 and CP43-Arg357) are included in the model system of the $Mn_4CaO_5$ cluster in water, $H_2O$ at W1 is not stable, releasing the proton, and is stabilized as $OH^-$ at W1 in the presence of protonated D1-Asp61 (Fig. 4a), i.e., $pK_a$(W1) << $pK_a$(W2) (Table 3). The absence of the corresponding acidic residue as an H-bond acceptor for W2 and the proceeding proton transfer pathway (e.g., the D1-Asp61 pathway for W1[26]) contribute to an increase in $pK_a$(W2) with respect to $pK_a$(W1).

In the isolated $Mn_4CaO_5$ cluster in water (in the absence of the PSII protein environment), proton release occurs along the transiently formed H-bond between the ligand water molecule and a bulk water molecule (i.e., mobile water molecule with a high dielectric constant $\approx 80$). The acceptor water molecule is best represented implicitly using the polarizable continuum model (PCM) method; in this case, the $pK_a$ value of the deprotonation site of the $Mn_4CaO_5$ cluster can be calculated, whereas the $pK_a$ difference between the deprotonation site and the adjacent proton-acceptor water molecule cannot be calculated directly. On the other hand, in the presence of the PSII protein environment, proton release occurs along the H-bond between the ligand water molecule and the fixed acceptor group (i.e., fixed dipole with a low dielectric constant << 80) (Fig. 4b). The acceptor group is represented explicitly based on the crystal structure; in this case, the energy barrier, which is associated with the $pK_a$ difference between the deprotonation site and the acceptor group[31], can be calculated based on the potential-energy profile of the H-bond, whereas the $pK_a$ value of the deprotonation site of the $Mn_4CaO_5$ cluster cannot be calculated directly. Note that only when the acceptor groups are always the same for all deprotonation sites (e.g., $H_2O$), the $pK_a$ values may be calculated from the $pK_a$ difference [e.g., the difference from $pK_a(H_2O/H_3O^+)$][31]. However, this is not the case for W1–W4 in the PSII protein environment, where the individual explicit acceptor groups already exist (e.g., D1-Asp61 for W1 and W446 for W2).

QM/MM calculations show that $H_2O$ at W1 forms a low-barrier H-bond with D1-Asp61 and the proton migrates towards the D1-Asp61 moiety in $S_2$ (Fig. 5a), whereas $H_2O$ at W2 forms a standard H-bond with an adjacent water molecule (W446, Fig. 4b) and the proton is localized at the W2 moiety, i.e., proton transfer from W2 to the acceptor $H_2O$ is energetically uphill (Fig. 5b). This suggests that $pK_a$(W1) is significantly lower than $pK_a$(W2) in the PSII protein electrostatic environment and W2 cannot release the proton as more deprotonatable W1 exists at the $Mn_4CaO_5$ moiety in the PSII protein environment[32]. The results are consistent with W2

**Table 2 $pK_a$ of the $Mn_4CaO_5$ cluster in water. Supplementary Table 2 for anti-ferromagnetic spin configuration.**

| S state | Mn1, Mn2, Mn3, Mn4[a] | W1 | W2 | W3 | W4 |
|---|---|---|---|---|---|
| $S_0$ | III, IV, III, III | 11.3 | 10.1 | 17.7 | 16.4 |
| $S_0$ [O4-H][b] | III, IV, III, III | 9.5 | 9.0 | 16.6 | 16.0 |
| $S_0$ [O5-H][c] | III, IV, III, III | 10.0 | 8.5 | 16.1 | 15.4 |
| $S_1$ | III, IV, IV, III | 10.2 | 9.0 | 15.6 | 15.1 |
| $S_2$ [open][d] | III, IV, IV, IV | 8.3 | 8.2 | 15.6 | 15.9 |
| $S_2$ [closed][e] | IV, IV, IV, III | 9.7 | 7.1 | 14.2 | 15.6 |

[a]Ferromagnetic spin configuration.
[b]O4 is protonated.
[c]O5 is protonated.
[d]Open-cubane structure.
[e]Closed-cubane structure.

**Table 3 $pK_a$ of the $Mn_4CaO_5$ cluster with D1-Asp61 and CP43-Arg357 in water.**

| S state | Mn1,Mn2,Mn3,Mn4[a] | W1 | W2 | W3 | W4 |
|---|---|---|---|---|---|
| $S_0$ | III, IV, III, III | <<$pK_a$(W2)[f] | 10.6 | 17.4 | 15.7 |
| $S_0$ [O4H][b] | III, IV, III, III | <<$pK_a$(W2)[f] | 9.6 | 15.5 | 14.8 |
| $S_0$ [O5H][c] | III, IV, III, III | <<$pK_a$(W2)[f] | 8.8 | 14.9 | 14.7 |
| $S_1$ | III, IV, IV, III | <<$pK_a$(W2)[f] | 10.2 | 16.6 | 13.6 |
| $S_2$ [open][d] | III, IV, IV, IV | <<$pK_a$(W2)[f] | 9.1 | 13.8 | 13.8 |
| $S_2$ [closed][e] | IV, IV, IV, III | <<$pK_a$(W2)[f] | 8.8 | 14.1 | 14.6 |

[a]Ferromagnetic spin configuration.
[b]O4 is protonated.
[c]O5 is protonated.
[d]Open-cubane structure.
[e]Closed-cubane structure.
[f]not determined because of deprotonation of W1 to D1-Asp61.

(a)                                        (b)

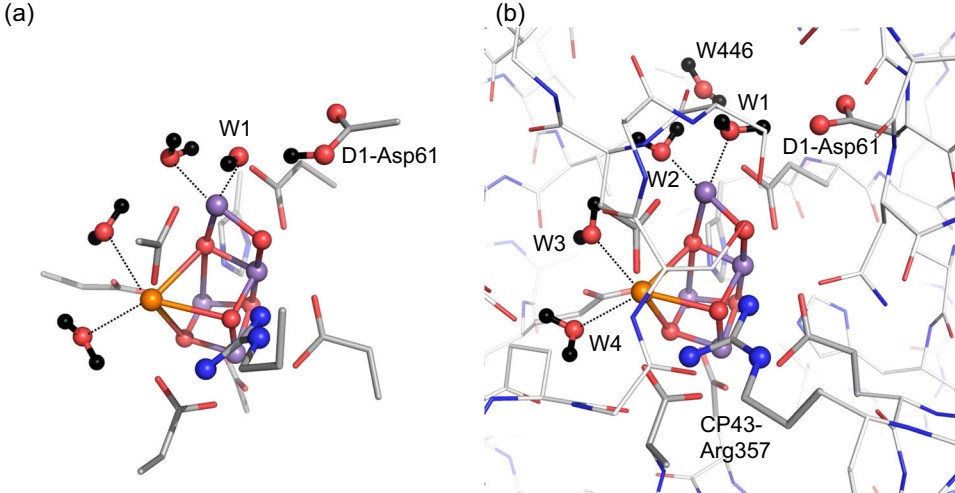

**Fig. 4 $Mn_4CaO_5$ structures including the second sphere ligand residues (D1-Asp61 and CP43-Arg357) in $S_1$.** Dotted lines indicate ligations of the ligand water molecules to metal ions. **a** Quantum-chemically optimized structure in the absence of the PSII protein environment. W1 is stabilized as $OH^-$ in the presence of protonated D1-Asp61, as the release of the proton occurs from $H_2O$ at W1 to ionized D1-Asp61. **b** QM/MM-optimized $Mn_4CaO_5$ structure in the PSII protein environment.

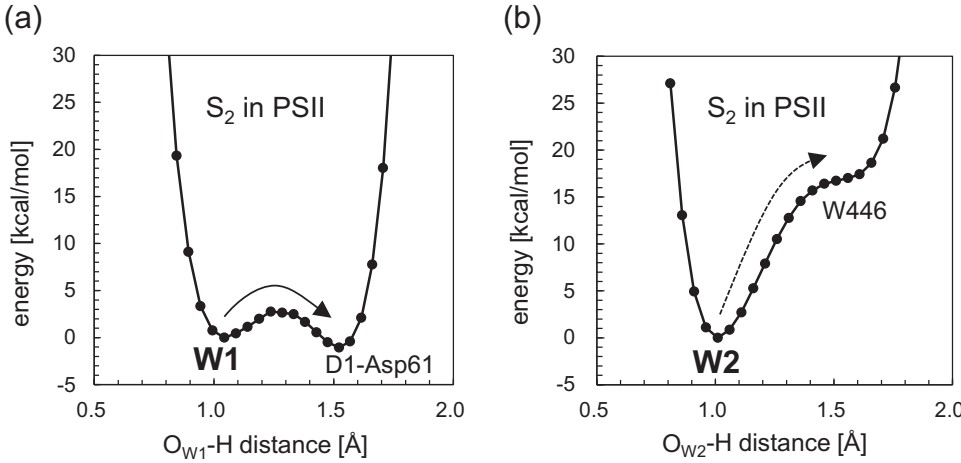

**Fig. 5 The potential energy profiles of the H-bonds in S₂ [(Mn1, Mn2, Mn3, Mn4) = (III, IV, IV, IV)] in the PSII protein environment. a** W1 and D1-Asp61. The isoenergetic proton transfer from W1 to D1-Asp61 indicates that p$K_a$(W1) ≈ p$K_a$(D1-Asp61). **b** W2 and the H-bond acceptor water molecule, W446. The energetically uphill proton transfer from W2 to W446 indicates p$K_a$(W2) >> p$K_a$(W1), i.e., deprotonation of W2 is energetically less favorable than deprotonation of W1.

being $H_2O$ in $S_1$ and $S_2$ based on FTIR spectra and theoretical calculations by Nakamura and Noguchi[8].

In summary, p$K_a$(W1) ≈ p$K_a$(W2) << p$K_a$(W3) ≈ p$K_a$(W4) in the $Mn_4CaO_5$ cluster in water (Table 2). p$K_a$(W2) is only marginally (~1 p$K_a$ unit) lower than p$K_a$(W1) in the absence of the PSII protein environment (Table 2) if the $Mn_4CaO_5$ cluster is defined as shown in Fig. 1, which may be a basis of why electron paramagnetic resonance (EPR) signals were often interpreted based on simplified theoretical models with $OH^-$ at W2 (e.g., refs. [10,22,23]). p$K_a$(W1) is significantly lower than p$K_a$(W2) if the $Mn_4CaO_5$ cluster includes the second sphere ligand residues, D1-Asp61 and CP43-Arg357 (Table 3, Fig. 4a). Thus, as p$K_a$(W1) and p$K_a$(W2) depend strongly on the definition of the $Mn_4CaO_5$ region, p$K_a$(W1) and p$K_a$(W2) in water (in the absence of the protein environment) do not provide any clue to understanding the deprotonation sites in the physiological S-state transitions. The potential energy profiles of the H-bonds show that in the presence of the PSII protein environment in $S_2$ Fig. 4b), $H_2O$ at W1 forms a low-barrier H-bond with D1-Asp61 and proton transfer is barrier-less (Fig. 5a)[26], whereas $H_2O$ at W2 forms a standard H-bond with the adjacent $H_2O$ and proton transfer is energetically uphill (Fig. 5b)[32]. These suggest that p$K_a$(W1) ≈ p$K_a$(D1-Asp61) << p$K_a$(W2) in PSII. As far as W1 exists, W2 can never release the proton (i.e., not $OH^-$) in PSII.

## Methods

**p$K_a$ calculation.** In the deprotonation reaction of the protonated state (AH) to deprotonated state (A⁻) in water, p$K_a$ is defined as

$$pK_a = \frac{\Delta G_{water}}{2.303\,RT}, \qquad (2)$$

where $\Delta G_{aq}$ is the free energy difference between (AH) and (A⁻ + H⁺) (i.e., $\Delta G_{water} = G_{water}(A^-) - G_{water}(AH) + G_{water}(H^+)$), $R$ is the gas constant, and $T$ is the temperature. $\Delta G_{water}$ can also be approximated as

$$\Delta G_{water} = k\Delta E_{water} + C, \qquad (3)$$

where $k$ is the scaling factor, $\Delta E_{water}$ is the energy difference between AH and A⁻, which can be calculated using a quantum chemical approach with the PCM method, and $C$ is the constant (simple p$K_a$ estimation with energy of the optimized geometry scheme[33]). If the p$K_a$ values of molecules are obtained at the same temperature, Eq. (2) can be written into Eq. (4) using the Eq. (3) as

$$pK_a = k'\Delta E_{water} + C', \qquad (4)$$

where $k'$ is the scaling factor and $C'$ is constant. To determine $k'$ and $C'$, we calculated $\Delta E_{water}$ for 23 hexa-aqua metal complexes whose experimentally measured p$K_a$ values are reported[21,34].

**Hexa-aqua metal complex.** The optimized geometry of the protonated hexa-aqua metal complex was obtained, using the restricted or unrestricted density functional theory with the B3LYP functional. The CSDZ* basis set was used for lanthanides except for La, the ERMLER2* basis set for actinides, and the LACVP* basis set for all other atoms. The spin states were consistent with previous studies by Galstyan et al.[34]. The optimized geometries of the deprotonated hexa-aqua metal complexes were obtained, fixing the torsion angles to prevent $OH^-$ from forming an H-bond with other ligand $H_2O$ molecules. However, the H-bond formation between the ligand $OH^-$ and $H_2O$ molecules could not be avoided for $Ca^{2+}$, $Zn^{2+}$, $Cd^{2+}$, $Dy^{3+}$, $Th^{4+}$, $Pa^{4+}$, $U^{4+}$, $Np^{4+}$, and $Pu^{4+}$, which were excluded from the present study. It should be noted that by adding a few external $H_2O$ molecules to the ligand $OH^-$ moiety, the H-bond formation between the ligand $OH^-$ and $H_2O$ molecules could be avoided without fixing the torsion angles. It was reported that the p$K_a$ values for hexa-aqua metal complexes in water did not differ significantly when calculated by fixing the torsion angles or adding a few external $H_2O$ molecules[34], probably because the shape of the hexa-aqua metal complex is symmetrical. However, this does not hold true for W1–W4 in the $Mn_4CaO_5$ cluster whose shape is not symmetric. Adding a few external $H_2O$ molecules to the ligand $OH^-$ moiety causes structural changes with respect to the original coordination geometry of the PSII crystal structure. In addition, explicit $H_2O$ water molecules (i.e., fixed dipole) form a specific $H_2O$ cluster (i.e., the dielectric constant << 80) at the deprotonatable ligand moiety, which does neither represent bulk water (i.e., the dielectric constant ≈ 80) nor provide the relevant p$K_a$ values. Based on these, the torsion angles were fixed to obtain the optimized geometry for p$K_a$ calculations in the present study. Using the optimized geometries, the energy difference ($\Delta E_{water}$) between the protonated and deprotonated states of hexa-aqua metal complexes were calculated with PCM method, using the Jaguar program[35].

**$Mn_4CaO_5$ cluster.** The optimized geometry of the $Mn_4CaO_5$ cluster in the PSII protein environment was obtained as follows: the atomic coordinates of PSII were taken from the X-ray structure of PSII monomer unit "A" of the PSII complexes from *Thermosynechococcus vulcanus* at a resolution of 1.9 Å (PDB code, 3ARC)[29]. Atomic partial charges of the amino acids were adopted from the all-atom CHARMM22[36] parameter set, respectively. D1-His337 was considered to be protonated[8]. We employed the electrostatic embedding QM/MM scheme, in which electrostatic and steric effects created by a protein environment were explicitly considered, and we used the Qsite[37] program code. We employed the unrestricted DFT method with the B3LYP functional and LACVP* basis sets. To analyze the $Mn_4CaO_5$ geometries and the H-bond potential-energy profiles, the QM region was defined as the $Mn_4CaO_5$ cluster (including the ligand side-chains of D1-Asp170, D1-Glu189, D1-His332, D1-Glu333, D1-Asp342, CP43-Glu354, the ligand carboxy-terminal group of D1-Ala344, and the ligand water molecules, W1–W4), the Cl-1 binding site (Cl-1, W442, W446, and the side-chains of D1-Asn181 and D2-Lys317), and the second-sphere ligands (side-chains of D1-Asp61 and CP43-Arg357). Specifically, the coordinates of the heavy atoms in the surrounding MM region were fixed at their original X-ray coordinates, while those of the H atoms in the MM region were optimized using the OPLS2005 force field. All of the atomic coordinates in the QM region were fully relaxed (i.e., not fixed) in the QM/MM calculation. All of the H-bond partners were included in the QM region. The cluster was considered to comprise ferromagnetically coupled Mn atoms, where the total spin $S = 15/2$ in $S_0$, $14/2$ in $S_1$, and $13/2$ in $S_2$. The resulting Mn oxidation states (Mn1, Mn2, Mn3, Mn4) were (III, IV, III, III) in $S_0$, (III, IV, IV, III) in $S_1$, (III, IV, IV, IV) in open-cubane $S_2$, and (IV, IV, IV, III) in closed-cubane $S_2$. It should be noted that the difference in $S$ (e.g., $S = 1/2$ in $S_2$[38], high,

low, ferromagnetic, and antiferromagnetic) did not affect the values; e.g., (i) the resulting geometry[39,40], (ii) the potential energy profile of proton transfer[26], (iii) the redox potential of each Mn site[41], and (iv) the p$K_a$ values for the ligand water molecules W1–W4 in the absence of the protein environment (see Supplementary Table 2). To obtain the potential energy profiles of the O…H$^+$…O bond, the QM/MM optimized geometry was used as the initial geometry. The H atom under investigation was moved between the two O moieties by 0.05 Å, after which the geometry was optimized by constraining the distance between O–H$^+$ and H$^+$–O distances, and the energy was calculated. This procedure was repeated until the H atom reached the O moieties.

In the absence of the PSII protein environment (i.e., in vacuum), the QM/MM-optimized geometry was re-optimized, using the unrestricted density functional theory with the B3LYP functional and LACVP* basis sets and fixing the torsion angles to maintain the overall shape of the less stable complex. The QM region was defined as either the $Mn_4CaO_5$ cluster (including the ligand side-chains of D1-Asp170, D1-Glu189, D1-His332, D1-Glu333, D1-Asp342, CP43-Glu354, the ligand carboxy-terminal group of D1-Ala344, and the ligand water molecules, W1–W4) or the $Mn_4CaO_5$ cluster and the second-sphere ligands (side-chains of D1-Asp61 and CP43-Arg357). Using the optimized geometries, the energy difference ($\Delta E_{water}$) between the protonated and deprotonated states of the $Mn_4CaO_5$ cluster were calculated with the PCM method, using the Jaguar program[35].

## Data availability

All data generated or analyzed during this study are included in this article (and its Supplementary Information files).

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

## Acknowledgements

This research was supported by JST CREST (JPMJCR1656 to H.I.), JSPS KAKENHI (18H05155, 18H01937, 20H03217, and 20H05090 to H.I., 16H06560 and 18H01186 to K. S.), and the Interdisciplinary Computational Science Program in CCS, University of Tsukuba (K.S.).

## Author contributions

H.I. designed research; K.S., M.N., and H.I. performed research; K.S., M.N., and H.I. analyzed data; and H.I. wrote the paper.

## Competing interests

The authors declare no competing interests.
