## [Peer Review File · Communications Chemistry]

Reviewers' comments:

Reviewer #1 (Remarks to the Author):

In the present manuscript, Saito et al. calculated the pKa values of four water molecules attached to the Mn₄CaO₅ cluster, which is the catalytic center of photosynthetic water oxidation in photosystem II. These water molecules are candidates of substrate and protons are released from some of them to form an O₂ molecule during the reaction cycle. Thus, estimation of pKa of the water molecules is essential in clarifying the molecular mechanism of water oxidation. They showed that the water molecules bound to Mn⁴⁺ (W1 and W2) have much lower pKa than those bound to Ca (W3 and W4), and the pKa of W1 is significantly low because of a strong hydrogen bond with a carboxylate side chain (D1-D61). Interestingly, the pKa of W2 is slightly lower than that of W1 in the isolated Mn₄CaO₅ cluster in the absence of the protein environment, providing the reason why the previous QM calculations of the Mn₄CaO₅ cluster preferred a deprotonated W2 structure. The calculated results in the present study are highly reliable and provide significant insight into the catalytic mechanism of water oxidation. The reviewer thus recommends the publication of this manuscript in Communications Chemistry.

Minor points to be addressed:

1. Figure 3 vs. Figure S1: Why does the plot become linear in water, whereas it is not in vacuum? The reason may be suggested.
2. The reason for the absence of the pKa table for the QM/MM calculation with a full protein environment may be briefly mentioned.
3. Table 1: The calculated pKa values may be obtained from eq. 1. This should be clarified. 4. Table S1: The calculated pKa values are identical to those in Table 1, but only RMSD is different. This seems strange.
5. line 40: observed -> observed
6. line 45: moiety 11-13).
7. line 96: Results and Discussion
8. line 137: Table 2 instead of Table 1.
9. line 139 and 140: W1 and W2 may be reverse.
10. line 241: "In the MM region,force fields." is the repeat of the sentence in line 239.
11. line 266: Insert a space before "It should..".

Reviewer #2 (Remarks to the Author):

The manuscript of Saito et al. is theoretical study of nature's water splitting cofactor, the Mn₄O₅Ca cluster found in Photosystem II, midway through its catalytic cycle – the S₂ state. The authors have focused on resolving the pKa's of the four water molecules which ligate the cluster: W1 and W2, bound to Mn⁴⁺ and W3 and W4, bound to the Ca²⁺ ion. As expected, they find that the pKa's of the two waters bound to the Ca²⁺ ion (W2, W3) are significantly higher than that of the Mn⁴⁺. More interestingly, they convincingly show that W1 and W2 have very different pKa's and that this difference is due to the protein environment: W1 and W2 form hydrogen bonds within the protein, but the interaction between W1 and Asp61 seems specifically tuned for proton egress.

I cannot fault the methods used in the study or its outcome - although whether its sensible to assign pKa values to a specific functional group within a hydrogen bonded network (such as a protein) is not immediately clear to me. That said, I don't believe it warrants publication in Nature Communications. The basic result – that proton egress from the catalyst likely proceeds via the Asp61 residue is well accepted in the literature, and has been characterized using sophisticated QM/MM approaches by a number of groups e.g. Guidoni (Narzi et al. 2014 PNAS), Pantazis (Retegan et al 2016 Chem Sci), Siegbahn (Siegbahn 2012 PCCP), Batista (Rivalta et al 2011 Biochemistry). There is also substantial experimental evidence, including mutagenesis studies of

the Asp61 (e.g. Debus et al. 2014 Biochemistry), that support this role. I suggest this work would be more appropriate for a technical journal.

Reviewer #3 (Remarks to the Author):

This paper reports an important and critical point for the mechanism study of OEC in PSII. The contribution by this study will be significant.

Taking H-bond networks into account, the possible OH in W1 or W2 with other Ws are examined. Evaluation of pKa values is carefully carried out. The arguments based on H-bond potential energy are also convincing. Contrasting with many previous theoretical studies which are often based on the total energy of limited Mn₄CaO₅ systems without sufficiently including ligands (eg. Asp61), this paper presents a sound approach. The reviewer recommends publication with minor revisions.

The reviewer requests some additional short deep discussion, reflecting a recent review by D.A.Pantazis (ACS, Catal,2019,8,9477), especially considering its section 4 which focuses on S₀ state.

Minor points,

1 line 217, fixing the torsion angles,,,,

To further confirm the validity of calculations, the reviewer proposes calculations with a few additional H₂O molecules on OH, without fixing geometries. It is possible to obtain free energy difference, by keeping mass balance.

2 line 137

Table 2, instead of Table 1 ?

Reviewer 1

The reviewer suggests several useful suggestions. We have modified the text to take into account the all of the reviewer's comments. The specific changes made are listed below.

1) *"Figure 3 vs. Figure S1: Why does the plot become linear in water, whereas it is not in vacuum? The reason may be suggested."*

→ In vacuum, electrostatic interaction between the metal valence (i.e., 1, 2, 3) and the ligand OH- (i.e., deprotonated state) is overestimated. This is why the tendency strongly depends on the net charge of 1, 2, and 3. We have mentioned in Results and Discussion (page 7, below eq. 1; Reviewer 1-1).

2) *"The reason for the absence of the pKa table for the QM/MM calculation with a full protein environment may be briefly mentioned."*

→ We have discussed in Results and Discussion (page 12, below Figure 5; Reviewer 1-2).

3) *"Table 1: The calculated pKa values may be obtained from eq. 1. This should be clarified."*

→ We have stated in the table legend (page 8, Table 1; Reviewer 1-3).

4) *"Table 1: The calculated pKa values may be obtained from eq. 1. This should be clarified. 4. Table S1: The calculated pKa values are identical to those in Table 1, but only RMSD is different. This seems strange."*

→ Table S1 was wrong. We have corrected it.

5) *"line 40: ovserved -> observed"*

→ We have corrected.

6) *"line 45: moiety 11-13)." "*

→ We have corrected.

7) *"line 96: Results and Discussion"*

→ We have corrected.

8) *"line 137: Table 2 instead of Table 1."*

→ We have corrected.

9) *"line 139 and 140: W1 and W2 may be reverse."*

→ We have corrected.

10) *“line 241: “In the MM region,force fields.” is the repeat of the sentence in line 239.”*

→ We have corrected (removed the “In the MM region...” sentence).

11) *“line 266: Insert a space before “It should..”.”*

→ We have corrected.

Reviewer 2

We have modified the text to take into account the all of the reviewer's comments. The specific changes made are listed below.

"I cannot fault the methods used in the study or its outcome - although whether its sensible to assign pKa values to a specific functional group within a hydrogen bonded network (such as a protein) is not immediately clear to me."

→ We never assign pKa values to a specific functional group within a hydrogen bonded network in a protein. We present pKa values only for the isolated cluster.

"That said, I don't believe it warrants publication in Nature Communications."

→ this is NOT Nature Communications.

"The basic result – that proton egress from the catalyst likely proceeds via the Asp61 residue is well accepted in the literature, and has been characterized using sophisticated QM/MM approaches by a number of groups e.g. Guidoni (Narzi et al. 2014 PNAS), Pantazis (Retegan et al 2016 Chem Sci), Siegbahn (Siegbahn 2012 PCCP), Batista (Rivalta et al 2011 Biochemistry). There is also substantial experimental evidence, including mutagenesis studies of the Asp61 (e.g. Debus et al. 2014 Biochemistry), that support this role.."

→ Asp61 is not the main focus in the present paper.

We have cited these papers (Siegbahn and others in page 3, 4, and 10, Debus in page 4).

The comment "sophisticated QM/MM approaches by a number of groups" is not correct. They are LESS sophisticated approaches, QM or MD.

- a) Siegbahn's models are simplified QM models, not QM/MM.
- b) The Pantazis one (Retegan et al 2016 Chem Sci, the same as Krewald et al. 2015 Chem Sci) is not QM/MM. It is just an oversimplified QM model (no protein environment). We have already pointed out, by illustrating in our recent paper (Saito and Ishikita. Biochim. Biophys. Acta 1860 (2019) 148059).
- c) The Batista paper (Rivalta et al 2011 Biochemistry) is neither QM nor QM/MM. It is just a classical MD study.

We have briefly mentioned in Results and Discussion (page 10; Reviewer 2)

Reviewer 3

The reviewer suggests several useful suggestions. We have modified the text to take into account the all of the reviewer's comments. The specific changes made are listed below.

“The reviewer requests some additional short deep discussion, reflecting a recent review by D.A.Pantazis (ACS, Catal,2019,8,9477), especially considering its section 4 which focuses on SO state.”

→ We have discussed in Introduction (page 3; Reviewer 3).

1) *“line 217, fixing the torsion angles,,,”*

To further confirm the validity of calculations, the reviewer proposes calculations with a few additional H₂O molecules on OH, without fixing geometries. It is possible to obtain free energy difference, by keeping mass balance.”

→ We have mentioned what happens when releasing the torsion angle (without adding a few water molecules) in Results and Discussion (page 9, below Table 2; Reviewer 3-1). We have also discussed in Methods (page 14, below Figure 5; Reviewer 3-1).

2) *“line 137 Table 2, instead of Table 1 ?”*

→ We have corrected.

REVIEWERS' COMMENTS:

Reviewer #1 (Remarks to the Author):

The manuscript was properly revised. The reviewer thus thinks that it is now in an acceptable form for publication.

Reviewer #3 (Remarks to the Author):

Editor's note: this reviewer offered no further comments to the authors.